# A Simple Model of Inference Scaling Laws

**Noam Levi** [1]

## Abstract

Neural scaling laws have garnered significant interest due to their ability to predict model performance as a function of increasing parameters, data, and compute. In this work, we propose a simple statistical ansatz based on memorization to study scaling laws in the context of inference. Specifically, how performance improves with multiple inference attempts. We explore the coverage, or pass@k metric, which measures the chance of success over repeated attempts and provide a motivation for the observed functional form of the inference scaling behavior of the coverage in large language models (LLMs) on reasoning tasks. We then define an "inference loss", which exhibits a power law decay as the number of trials increases, and connect this result with prompting costs. We further test the universality of our construction by conducting experiments on a simple generative model, and find that our predictions are in agreement with the empirical coverage curves in a controlled setting. Our simple framework sets the ground for incorporating inference scaling with other known scaling laws.

## 1. Introduction

Advancements in deep learning have demonstrated that the performance of neural networks scales predictably as a function of model size, dataset size, and computational resources (Hestness et al., 2017; Kaplan et al., 2020a; Rosenfeld et al., 2020; Henighan et al., 2020a). These trends, known colloquially as *neural scaling laws*, have motivated research into understanding how scaling influences model performance in a wide range of domains, but in particular Large Language Models (LLMs) (Brown et al., 2020; Hoffmann et al., 2022).

[1]École Polytechnique Fédérale de Lausanne (EPFL), Lausanne, Switzerland. Correspondence to: Noam Levi <noam.levi@epfl.ch>.

*Proceedings of the 42nd International Conference on Machine Learning*, Vancouver, Canada. PMLR 267, 2025. Copyright 2025 by the author(s).

While the scaling laws literature focuses on test performance induced by pre-training, scaling during *inference*—the process by which a trained model makes predictions on new data—has received less attention.

Recent works have shown empirically that LLMs can gain substantial benefits from repeated prompts to perform better on difficult tasks such as coding and formal proofs, where verification of the correct answer can be done (Brown et al., 2024; Snell et al., 2024; Bansal et al., 2024). These works demonstrate that the performance of weaker models can be amplified without further training by repeating inference trials, giving rise to a natural question:

*Can we interpret, or predict the inference scaling behavior of a model with repeated attempts?*

To answer this question, we propose a simple toy model that isolates the *inference scaling laws* which dictate how certain performance metrics improve as a function of the number of inference attempts.

Inspired by the work of Hutter (2021), which introduced a model to study scaling behavior for memorization and generalization capabilities, we devise a simple setting to capture the effect of repeated inference attempts, focusing on the *coverage* metric, also known as *pass@k*.

In this work, we present analytical predictions for coverage from a probabilistic perspective and demonstrate how inference improves with the number of repeated trials in a predictable way, which matches the observed behavior in (Brown et al., 2024) and (Snell et al., 2024).

We use two different approaches to obtain the predicted pass@k, and highlight the connection between coverage and total inference cost. Additionally, we define a simple "inference loss", similar to the familiar test loss, but allowing for repeated trials, and predict its scaling.

Our main predictions are verified by empirical results on mathematical reasoning tasks for several LLMs, following Brown et al. (2024).

Lastly, we test the universality of our theory on an entirely different generative model. We train a Variational Autoencoder (VAE) (Kingma and Welling, 2022) to generate reconstructions of its training data by sampling from a latent

space with an associated temperature. We find that the same behavior persists for both LLMs and the VAE setup, in spite of the vast differences in models and tasks.

Given that our results are isolated from the effects of other neural scaling laws, they could be incorporated into a broader exploration to find the optimal train/inference point. In particular, we hope this work sets the ground for exploring the optimal trade-off between training and inference attempts, such that the total performance is improved while cost is minimized.

The rest of the paper is organized as follows: In Section 3, we discuss our main setup, analogizing large models to perfect memorizers. We explain how this model can lead to the empirical pass@k curves for LLMs in Section 4, and provide an interpretation for the parameters of the model as defining an effective perceived difficulty for the dataset. We further connect these parameters with compute costs in Section 4.3. In Section 5, we reaffirm our results using a controlled simple generative model. We conclude in Section 6 and discuss future directions.

## 2. Related Work

**Neural Scaling Laws:** Scaling laws for neural networks have been extensively studied in recent years. Empirical research has shown that error rates decrease predictably as a function of increased data, parameters, and compute, following power-law relationships. Notable contributions in this space include work by Kaplan et al. (2020b), who demonstrated consistent scaling behavior across language models, and Henighan et al. (2020b), who extended these ideas to multimodal models, as well as Cabannes et al. (2024); Maloney et al. (2022); Bordelon et al. (2020); Spigler et al. (2020); Caponnetto and De Vito (2007); Steinwart et al. (2009); Fischer and Steinwart (2020); Cui et al. (2021); Levi and Oz (2023; 2024); Nam et al. (2024) who studied scaling laws for solvable yet sufficiently complex models, ranging from generalized linear regression on random feature models to kernel ridge regression, connecting them to results from random matrix theory and the underlying properties of the data.

While most scaling laws focus on training, the study of *inference scaling* remains under-explored. Our work strives to fill this gap by studying how performance improves with repeated inference attempts.

**Inference and reasoning:** Recent advancements in inference scaling and reasoning within LLMs have been significantly influenced by techniques such as Chain-of-Thought (CoT) prompting and Tree-of-Thought (ToT) reasoning. CoT prompting, as explored by (Wei et al., 2022), enables models to generate intermediate reasoning steps, enhancing their problem-solving capabilities. Building upon this,

(Yao et al., 2023) introduced ToT reasoning, which scales the CoT approach by structuring reasoning paths as trees, allowing for exploration of multiple reasoning paths simultaneously. These methods draw inspiration from earlier works like AlphaGo (Silver et al., 2016) and AlphaZero (Silver et al., 2017), which demonstrated the effectiveness of self-play and guided self-trajectory training in complex decision-making tasks. Additionally, the concept of self-consistency, as discussed by (Wang et al., 2022), employs majority voting over multiple reasoning paths to improve answer accuracy. These collective efforts aim to enhance the reasoning abilities of LLMs by refining their inference processes and scaling their reasoning capabilities, potentially leading to scaling laws for performance (Wu et al., 2024). However, the way these inference processes scale with the number of inference attempts is poorly understood.

## 3. Memorizing Ansatz

In the following, we first briefly review the simplest model which produces the known data scaling law prediction by appealing to a memorizing construction, then consider our proposed model for repeated inference attempts.

The *Hutter model* (Hutter, 2021) is a probabilistic framework originally introduced to study the scaling behavior of learning curves, focusing on the relationship between memorization and generalization. It assumes a model which perfectly memorizes features during training, allowing it to correctly match sets of features and labels $\{i, y_i\}$, such that only unseen features can incur an error. The set of features is assumed to follow a Zipf power law decay with parameter $\alpha$, where the probability of encountering feature $i$ is $\theta_i \propto i^{-1-\alpha}$ and decreases with its rank. For $n$ training samples, the expected single feature error $E_i$ is

$$\mathcal{E}_n = \mathbb{E}[E_i] = \sum_{i=1}^{\infty} \theta_i (1-\theta_i)^n \approx n^{-\beta}, \quad \beta = \frac{1}{1+\alpha}. \quad (1)$$

Equation (1) captures the average likelihood of encountering and labeling the feature incorrectly after $n$ training samples. Similar to this model, we will adopt the idea of memorization as a surrogate for training, but will depart from the finite training set assumption as a basis for test errors.

### 3.1. Perfect Sample Memorization, Imperfect Predictions

In contrast to the Hutter model, we focus on a scenario where *all samples up to the model capacity $n_c$ have been memorized*, hence there is no notion of test error coming from unseen data. Instead, failure during inference or data generation could arise from the need to follow a sequence of steps to reach the correct answer.

Concretely, we consider a joint model of memory $M$ and in-

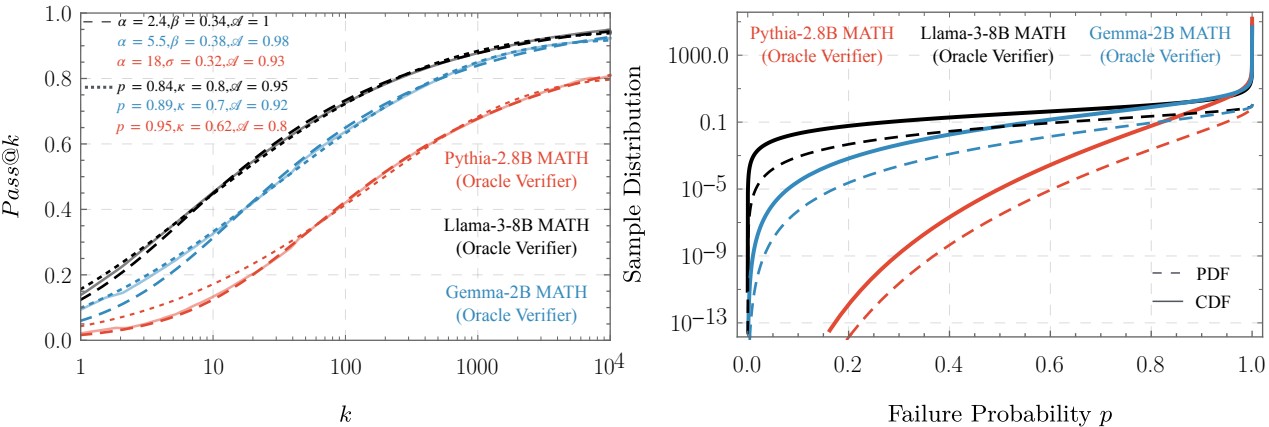

*Figure 1.* **Pass@k and failure distribution curves for various LLMs on difficult tasks, against theoretical scaling predictions**. *Left:* The relationship between pass@k and the number of samples for several coding and maths tasks for different models, as described in (Brown et al., 2024), compared with the analytical predictions presented in Equations (7) and (13). The *solid* curves are data, while the *dashed* curves are the predictions from Equation (7), where $\alpha$ and $\beta$ correspond to the concentration of easy and hard problems, respectively. The *dotted* curves are the results of Equation (13). The functional form in both cases captures well the LLM pass@k curves for various models, by adjusting $\alpha, \beta$ or $p, \kappa$. *Right:* The Beta$(\alpha, \beta)$ distributions for the failure probabilities are shown for the different models. We can see that most of the questions are "difficult", while the left tail behavior informs us regarding the rate of improvement with additional trials on the "easier" samples. For example, the right panel demonstrates that Pythia-2.8B perceives the MATH dataset as having a larger density of difficult questions (PDF concentrated at $p_i$ close to 1), while Llama3-8B finds the same dataset easier.

ference $I$. The memory is $M \in \mathcal{M} := \mathcal{X} \rightarrow \mathcal{Y}$, where the samples are identical to the labels $\{x_i\}_{i=1}^n = \{y_i\}_{i=1}^n$, and corresponds to a model which simply learns to memorize its training data. The inference model $I \in \mathcal{H} := \mathbb{N} \rightarrow M$ takes an input index $i$, which serves as a proxy for a particular prompt or task, and should produce the associated memorized label (sample) $y_i$ by recalling it from the memory $M$. The inference model is taken to be imperfect in its retrieval, and subject to some error $\epsilon$, it makes predictions

$$I(i) = \begin{cases} y_i + \epsilon, & \text{with probability } p_i, \\ y_i, & \text{with probability } 1 - p_i, \end{cases} \quad (2)$$

for any sample $i = 1, \ldots, n_c$. For samples outside the model capacity the prediction is always wrong $I(i) \neq y_i$.

At this point, we only accept perfect model "answers", such that the performance of the model on a single sample is measured simply by

$$A(i) = \mathbf{1}_{\{y_i\}}(I(i)), \quad (3)$$

where $\mathbf{1}_{\mathcal{Y}}(x) = 1$ if $x \in \mathcal{Y}$ else 0 is the indicator function.

We start from the general case, where each sample has an unknown failure probability $p_i \in [0, 1]$ at inference, and we are interested in the probability of at least one successful generation of a sample over $k$ attempts. This is in-line with the type of reasoning tasks studied by (Snell et al., 2024; Brown et al., 2024), where the performance of a model requires only one correct answer, rather than every answer

to be correct.

The rest of our analysis relies on the following assumptions:

**Assumption 3.1.** For every sample $i$, we have access to a perfect verification method, which can determine if there exists a correct generated answer during inference $I(i) = y_i$, among $k$ possible candidates $\{I_1(i), \ldots, I_k(i)\}$.

**Assumption 3.2.** Inference attempts $\{I_1(i), \ldots, I_k(i)\}$ are independent and identically distributed (i.i.d.) random variables.

We note that Assumption 3.1 sets aside the important topic of the quality of answer verification. In tasks such as coding and formal proofs, automatic verification is often possible, as a candidate solution can be quickly identified to be correct by proof checkers (Zheng et al., 2021). Likewise, unit tests can be used to verify the correctness of candidate solutions to coding tasks (Jimenez et al., 2024). How the following analysis changes if we relax this assumption to the case of an imperfect verification method is an interesting question that we postpone to future studies.

Under Assumption 3.2, the probability of the model failing on all $k$ trials for a sample within its capacity with failure probability $p_i$ is simply

$$\mathbb{P}(k \text{ failures on sample } i) = \prod_{t=1}^{k} (1 - A_t(i)) = p_i^k, \quad (4)$$

where $t$ is the trial index. Therefore, the probability of at least one success in $k$ trials averaged over the entire dataset of size $n$, known as **pass@k**, is given by

$$\text{pass@k} = \frac{n_c}{n} \times \left( 1 - \frac{1}{n_c} \sum_{i=1}^{n_c} \prod_{t=1}^{k} (1 - A_t(i)) \right) \quad (5)$$

$$= \mathcal{A} \times \left( 1 - \frac{1}{n_c} \sum_{i=1}^{n_c} p_i^k \right),$$

where we define the fraction of samples the model can store as $\mathcal{A} = n_c/n \in [0,1]$, describing the maximal potential pass@k for a given model. This will be the metric we use to describe inference accuracy for the rest of this work.

# 4. Power Law Distributions Predict Inference Scaling for LLMs

The setup described in Section 3 is completely specified by the distribution of failure probabilities $p_i$. To construct the failure distribution, we assume that different samples may have different inference complexity levels, incorporating some "easy" and some "difficult" samples with respect to the inference model.

One way to model the different complexities is to appeal to the so-called Beta distribution. We think of the failure probability across samples itself $p = p_i$ as a random variable, drawn from $p \sim \text{Beta}(\alpha, \beta)$, whose probability density function (PDF) is given explicitly by

$$\text{Beta}(\alpha, \beta; p) = \frac{p^{-1+\alpha}(1-p)^{-1+\beta}}{B(\alpha, \beta)}, \quad (6)$$

where $B(\alpha, \beta)$ is the Euler beta function.

The $\text{Beta}(\alpha, \beta)$ distribution parameters are $\alpha \in \mathbb{R}^+$, which controls the amount of "easier" problems in the sample dataset, where smaller $\alpha$ pushes the distribution mass towards zero, while $\beta \in \mathbb{R}^+$ dictates how often we encounter "harder" problems. Namely, a lower $\beta$ parameter increases the distribution mass towards the right tail (high failure probabilities).

We can therefore compute the pass@k metric by simply averaging over the failure distributions as

$$\text{pass@k} = \mathcal{A} \times \left( 1 - \frac{1}{n} \sum_{i=1}^{n} p_i^k \right) \approx \mathcal{A} \times \left( 1 - \langle p^k \rangle \right)$$

$$= \mathcal{A} \times \left( 1 - \int_0^1 dp \, p^k \frac{p^{-1+\alpha}(1-p)^{-1+\beta}}{B(\alpha, \beta)} \right)$$

$$= \mathcal{A} \times \left( 1 - \frac{\Gamma(\beta)\Gamma(k+\alpha)}{B(\alpha, \beta)\Gamma(k+\alpha+\beta)} \right),$$
$$(7)$$

where $\Gamma(z)$ is the Euler gamma function, and $\langle . \rangle$ indicates

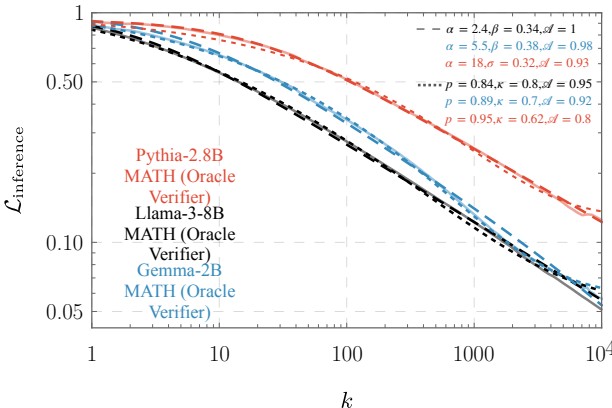

Figure 2. **Inference attempts loss $\mathcal{L}_{\text{inference}}(k)$ for repeated attempts on mathematics tasks**. The inference loss as a function of trials for the LLM experiments in (Brown et al., 2024).

averaging over the $p$ distribution.

To test these predictions, we utilize the reported pass@k results of (Brown et al., 2024), which evaluated Gemma-2B (Team et al., 2024), Llama3-8B (AI@Meta, 2024) and Pythia-2.8B (Biderman et al., 2023) on mathematical and coding tasks. Here, we take the results for the MATH dataset, which consists of difficult math word problems (Chen et al., 2024), where 128 random problems from the test set were chosen for evaluation.

In Figure 1, we show that the functional form of pass@k given in Equation (7) is a good approximation for the empirical pass@k curves for the LLMs evaluated on mathematical tasks, as reported in Brown et al. (2024).

At large $k$ values, we find a power-law decay of the effective average failure probability

$$\text{pass@k} \underset{k \to \infty}{\approx} \mathcal{A} \times \left( 1 - \frac{\Gamma(\beta)k^{-\beta}}{B(\alpha, \beta)} \right), \quad (8)$$

which is a common feature in neural scaling laws (Bahri et al., 2024). If we then define the **inference loss $\mathcal{L}_{\text{inference}}(k)$** as the expectation with respect to the sample distribution over errors, our results for pass@$k$ correspond to

$$\mathcal{L}_{\text{inference}}(k) \equiv \mathbb{E}(\text{Error in } k \text{ trials}) = \mathbb{E}(\mathcal{A} \times p^k) \quad (9)$$

$$\approx \mathcal{A} \times \frac{\Gamma(\beta)\Gamma(k+\alpha)}{B(\alpha, \beta)\Gamma(k+\alpha+\beta)}$$

$$\underset{k \to \infty}{\approx} \mathcal{A} \times \frac{\Gamma(\beta)k^{-\beta}}{B(\alpha, \beta)}.$$

This result implies that the model test loss with repeated inference steps will decrease mainly depending on the value of the exponent $\beta$. Intuitively, it means that for a fixed $\alpha$ parameter, the harder the questions appear to the model, the more inference attempts are required to reach a low loss. We illustrate this behavior in Figure 3.

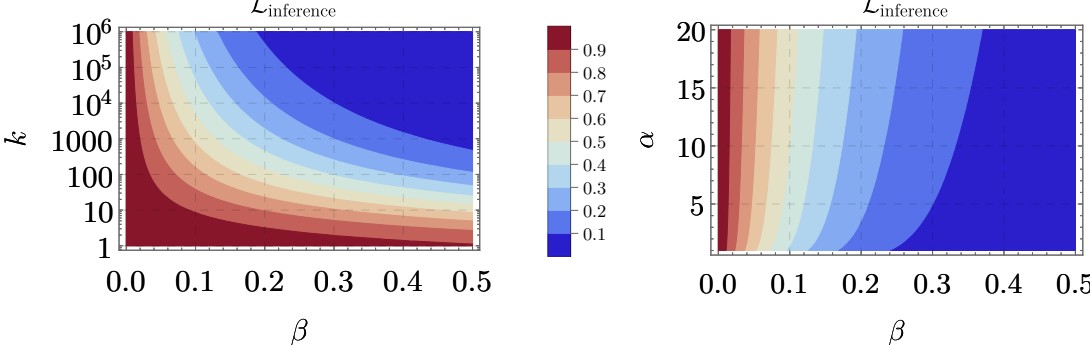

Figure 3. **Inference attempts loss $\mathcal{L}_{\text{inference}}(k)$ for repeated attempts on the memorizing model for different parameter choices**. Inference loss for different $\beta$ and $k$ values. Different colors indicate inference loss values at fixed $\alpha = 5$ (*left*) and at fixed $k = 10^4$ (*right*), illustrating the behavior of Equation (9).

## 4.1. Interpretability

The result in Equation (9) not only allows one to predict a threshold for inference in order to get an increase in model performance on difficult problems, but could also offer some interpretation for the difficulty of tasks with respect to the trained model.

We can gain some insights regarding the real-world models by fitting the pass@k metric according to Equation (7) to the ones given in (Brown et al., 2024), and attempt to interpret the properties of the test data from the parameters $\alpha$ and $\beta$. In particular, note that the ratio $\frac{\alpha}{\alpha+\beta}$ gives the mean of the Beta distribution, which represents the average failure probability across the samples. If $\alpha > \beta$, the mean failure probability is high (i.e., most samples are harder). On the other hand, if $\alpha < \beta$, the mean failure probability is low, implying that most samples are easy. Furthermore, the denominator $\alpha + \beta$ governs the concentration of the distribution, where a large $\alpha + \beta$ means that the failure probabilities $p_i$ are more tightly clustered around the mean (more homogeneity in difficulty).

From Figure 1, we can see that the typical values for the "harder" problem parameter $\beta \sim 0.35$, while the tail parameter is $2 < \alpha < 20$. This implies that most of the problems in the datasets used to measure the pass@k curves were indeed difficult in terms of the model's perception, while the existence of a left tail implies that the easy samples are covered quickly with less than 100 trials. This is reflected in the right panel of Figure 1.

An additional point of interest would be connecting the observed empirical pass@k curves with the unknown failure probability distribution of the different models. This can be done by noting that the average $\langle p^k \rangle$, performed in Equation (7), can be thought of as performing a Laplace transform from the failure probability variable $\sigma = \log(1/p)$ to

the trials space $k$,

$$
\begin{aligned}
\tilde{f}(k) = \langle p^k \rangle &= \int_0^\infty d\sigma e^{-\sigma k} \frac{e^{-\alpha\sigma}(1 - e^{-\sigma})^{-1+\beta}}{B(\alpha, \beta)} \\
&= \int_0^\infty d\sigma e^{-\sigma k} f(\sigma).
\end{aligned} \tag{10}
$$

This interpretation implies that it is possible to derive the probability distribution function of the samples in terms of their perceived difficulty by performing the inverse transform. In particular, given an empirical pass@k metric obtained for a given model, the inverse transform on $\tilde{f}(k) = (\mathcal{A} - \text{pass@k})/\mathcal{A}$ will yield the perceived difficulty PDF. Potentially, such a procedure can be used to identify "difficult" and "easy" questions and construct improved fine-tuning algorithms by choosing training samples biased towards the "difficult" but obtainable tasks.

## 4.2. Correlated Trials and Effective $k$ Approach

In the previous section, we showed that correlated samples drawn i.i.d. from a varying complexity distribution can effectively describe the pass@k metric for memorizing models. Here, we take a converse approach, where do not assume that samples are correlated through their failure rate distribution, but instead that *trials* themselves are correlated.

One can conjecture that dependencies between trials arise due to the internal model structure and the data itself. To capture these correlations, we suggest a model where the correlation between trials decays as a power law, implying that successive trials become less independent as we increase the number of trials.

In order to incorporate the correlation between trials, we define the notion of an *effective number of independent trials*, denoted $k_{\text{eff}}$. This adjusts the original $k$ to account for the decay in trial independence. The correlation between trials is modeled via a power-law decay in the eigenvalues

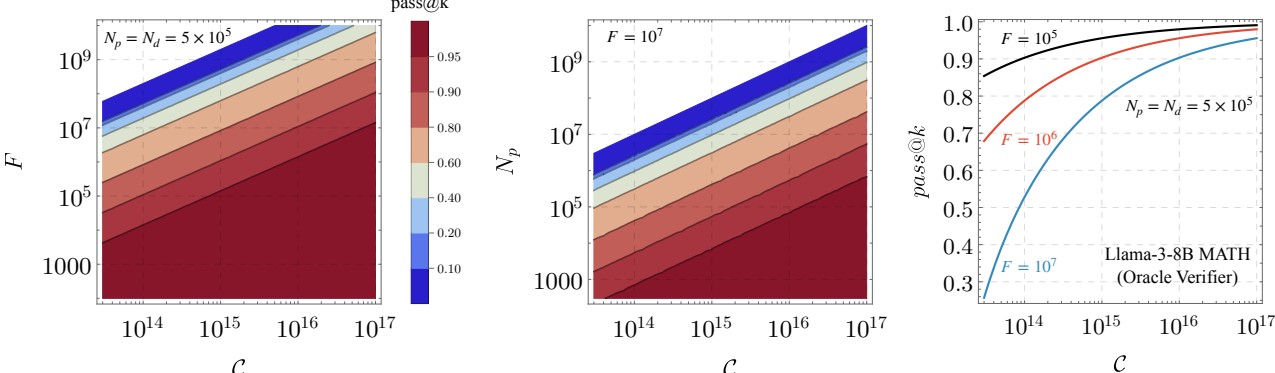

*Figure 4.* **Pass@k as a function of total inference cost for Llama-3-8B MATH (Oracle Verifier).** *Left and Center:* We show the pass@k metric as a function of number of total inference cost and number of FLOPS per token $F$ or number of prompt/decode tokens $N_p = N_d$ in log, log. We see that there is a clear trade-off between total inference cost whenever keeping one of the parameters fixed, in a predictable way from Equation (16). *Right:* We show a slice of the contour plots for fixed $N_p = N_d$, and changing the number of FLOPS per token. The parameters chosen for these figures are fitted from Equation (7) applied to the data taken from (Brown et al., 2024).

of the correlation matrix, such that the effective number of independent trials is given by

$$k_{\text{eff}} = \sum_{i=1}^{k} i^{-\kappa} = H_k(\kappa), \qquad \kappa \in \mathbb{R}_{\geq 0}, \qquad (11)$$

where $\kappa$ is the power-law exponent governing how quickly the correlation between trials decays, $H_k(\kappa)$ is the Harmonic number, and $\zeta(\kappa)$ is the Riemann zeta function. At the large inference trial limit, Equation (11) asymptotes to

$$k_{\text{eff}} \underset{k \to \infty}{\approx} \left( \frac{1}{2} - \frac{k}{\kappa - 1} \right) k^{-\kappa} + \zeta(\kappa) \qquad (12)$$

Thus, the probability of at least one success in $k$ trials, incorporating correlations, becomes

$$\text{pass@k} = \mathcal{A} \times \left( 1 - \frac{1}{n} \sum_{i=1}^{n} p_i^{k_{\text{eff}}} \right) = \mathcal{A} \times \left( 1 - p^{H_k(\kappa)} \right) \tag{13}$$

$$\approx \mathcal{A} \times \left( 1 - p^{\left( \frac{1}{2} - \frac{k}{\kappa-1} \right) k^{-\kappa} + \zeta(\kappa)} \right),$$

where $p = p_i$ is the error probability of every sample, and $k_{\text{eff}}$ accounts for correlations between trials. The result of Equation (13) is shown in Figure 1 as the dotted curves, which approximate the LLM behavior well for $k \gg 1$. We stress that this approach should be taken as an effective description, which nevertheless manages to accurately capture the same behavior as sample correlations.

## 4.3. Connection to Compute Scaling

Here, we would like to translate our results from the attempts variable to the inference cost. A natural proxy for the inference cost may be the number of required Floating Point Operations Per Second (FLOPS)[1]. For concreteness, we adapt the total inference cost formula suggested in (Brown et al., 2024), given by

$$\mathcal{C} = N_p \times F + N_d \times F \times k, \tag{14}$$

where $\mathcal{C}$ is the total inference cost, $N_p, N_d$ are the number of prompt and decode tokens, respectively, and $F$ is the number of FLOPS per token.

We can convert some of our pass@k results to this metric by taking the large $k$ limit of Equation (7), giving

$$k = \left( \frac{(\mathcal{A} - \text{pass@k}) B(\alpha, \beta)}{\mathcal{A} \Gamma(\beta)} \right)^{-1/\beta}, \tag{15}$$

hence the resulting coverage is simply

$$\text{Coverage}(\mathcal{C}) \approx \mathcal{A} \cdot \left( 1 - \frac{\Gamma(\beta)}{B(\alpha, \beta)} \left( \frac{\bar{\mathcal{C}} - N_p}{N_d} \right)^{-\beta} \right), \tag{16}$$

where $\bar{\mathcal{C}} \equiv \mathcal{C}/F$ is the normalized total inference cost. Keeping all uncontrollable parameters fixed, such as the number of FLOPS per token fixed, we see from Figure 4 that it is worthwhile to reduce the number of prompting and decoding tokens or increase the number of attempts, up to

---

[1]This may be too crude a cost metric that ignores other aspects of system efficiency (Dehghani et al., 2021), which may benefit from repeated sampling (Juravsky et al., 2024; Zheng et al., 2023), as pointed out in (Brown et al., 2024).

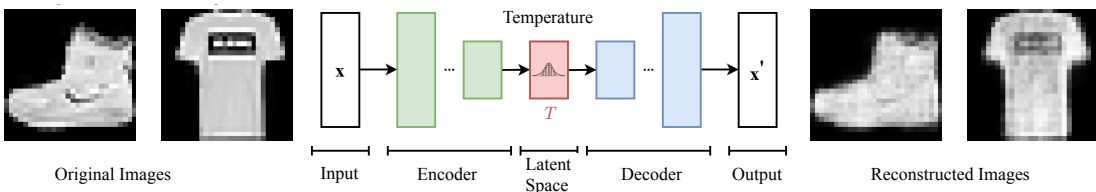

Temperature

Original Images    Input    Encoder    Latent Space    Decoder    Output    Reconstructed Images

*Figure 5.* **Visualization of the task described in Section 5**. Here, a VAE is tasked with generating samples from its training data, where a "failure" occurs when the reconstruction error falls above a certain threshold $\epsilon$.

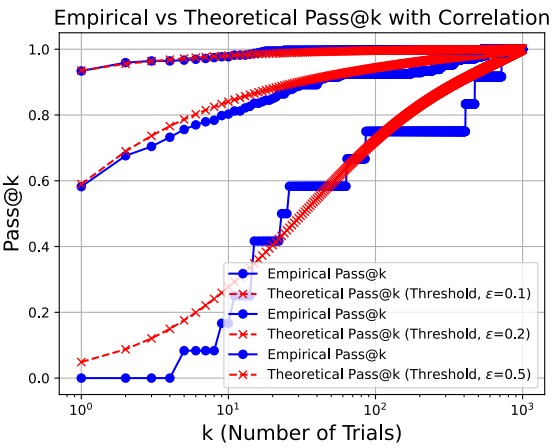
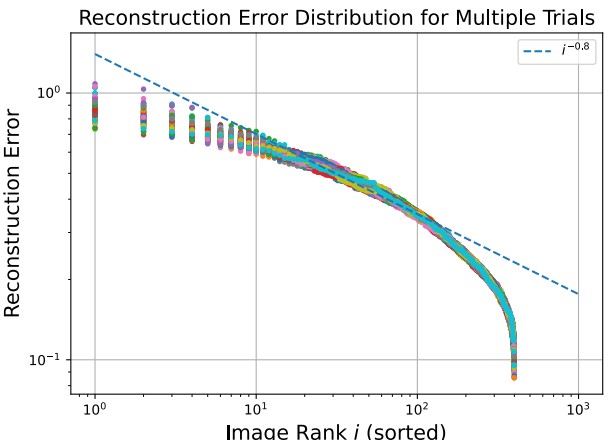

*Figure 6.* **Results for the VAE reconstruction task, compared with semi-analytical predictions**. *Left:* The pass@k metric as a function of number of attempts $k$, for different threshold values, with temperature $T = 1.1$. The curves have been normalized to asymptote at 1 for visual clarity. *Right:* The reconstruction error behavior across multiple trials, indicated by different colors. The errors obey a quasi power law behavior.

the minimal amount of total inference compute required to reach a target coverage. The numbers in Figure 4 are chosen to mimic the results found in (Brown et al., 2024), and are only meant to give an illustration of the functional behavior of Equation (16).

Alternatively, we can phrase Equation (16) in terms of the inference loss as

$$\mathcal{L}_{\text{inference}}(\mathcal{C}) \approx \mathcal{A} \times \frac{\Gamma(\beta)}{B(\alpha,\beta)} \left( \frac{\bar{\mathcal{C}} - N_p}{N_d} \right)^{-\beta}, \qquad (17)$$

which demonstrates the power law decay of the inference loss with total inference cost, depending on the value of $\beta$.

## 5. Experiments on a Simple Generative Model

To further validate some of our analytical understanding in a controllable setting, we perform a series of experiments in which we train a simple generative model to reconstruct images taken from Fashion-MNIST (Xiao et al., 2017). Our goal is to connect the theoretical memorizing model and the behavior of more complex generative models by attempting

to accurately reconstruct "memorized" examples, heuristically shown in Figure 5.

To do this, we train a VAE with a temperature parameter to study how errors propagate over multiple trials and to compare empirical pass@k with theoretical predictions under correlated trials. We refer to this as the *VAE reconstruction task*.

To quantify the error probability of the model over multiple samples, we define the error per sample using the norm of the difference between the reconstructed and original image:

$$\text{error}(i) = \frac{\|\hat{y}_i - y_i\|}{\|y_i\|}. \qquad (18)$$

Here, $y_i$ represents the original image, and $\hat{y}_i$ is the reconstruction. This per-sample error metric allows us to define success or failure at the sample level, where a trial is considered successful if the reconstruction error falls below a threshold $\epsilon$.

To empirically calculate pass@k, we sample multiple reconstructions from the VAE for each input sample. For each

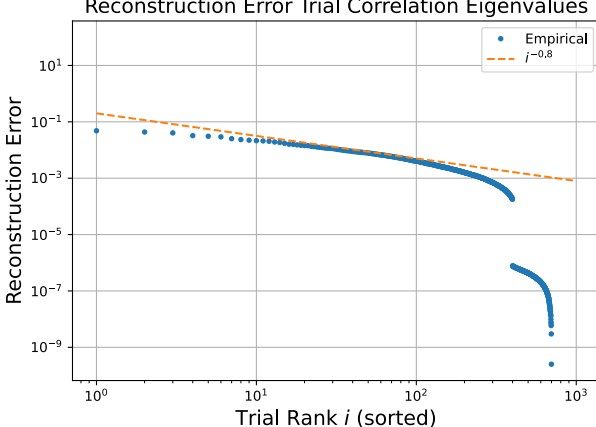

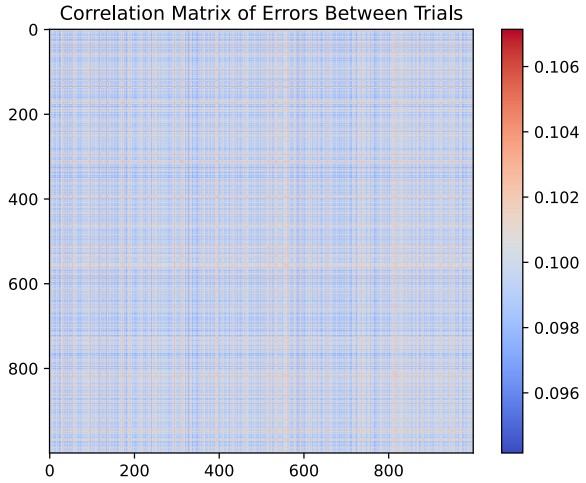

*Figure 7.* **Correlation matrix for errors across different trials for the VAE reconstruction task.** *Top:* The eigenvalues of the correlation matrix follow a power law decay with the number of trials. *Bottom:* Visualization of the correlation matrix itself shows clear correlations between different trials for $k = 1000$.

sample, we conduct $k$ trials and calculate whether at least one trial resulted in a reconstruction with error less than some chosen threshold value $\epsilon \in [0, 1]$. Pass@k is then computed as the fraction of samples for which the model succeeded to reconstruct at least once in $k$ trials.

In Figure 6, we show the pass@k results for the VAE reconstruction task, for different threshold $\epsilon$ values (left), as well as the reconstruction error distribution for multiple attempts (right). The theoretical predictions for the pass@k curves are shown for the effective $k$ approach, given in Equation (13), and approximate the VAE results well. Here, $p_i$ is computed empirically by taking the distribution at the maximal $k$ and $\kappa$ is taken from the correlation matrix in Figure 7.

To complete the picture, we empirically confirm that the assumption of independent trials is indeed violated, as trials are effectively correlated. To capture this effect, we compute the correlation matrix for the errors across trials $\epsilon_{kk'}$ as

$$\epsilon_{kk'} = \frac{1}{n} \sum_i \text{error}_{i,k} \times \text{error}_{i,k'}. \tag{19}$$

The eigenvalues of the correlation matrix decay as a power law, suggesting that the effective number of independent trials diminishes as $k$ increases, which is clearly depicted in Figure 7 (left).

## 6. Conclusions

In this paper, we have proposed a simple statistical explanation for a so-called inference scaling law, which describes the scaling of the coverage (pass@$k$) metric with number of repeated attempts for generative models. We presented two possible models which lead to inference scaling: One based on introducing a sample space distribution of "easy" and "difficult" problems, and the other on an effective Zipf-like correlation structure between trials. Using these simple constructions, we were able to derive analytical predictions for the pass@k, as well the test loss as a function of repeated attempts, which we dubbed the inference loss. We then verified our predictions empirically both through previous experimental results for LLMs and for a simple generative VAE construction.

We stress that the merit of our construction is in its simplicity, and there are many other models who can give rise to the same functional behavior. We view this as a positive rather than a negative, since it means that this simple model captures a universal behavior, which should not depend much on the modeling itself. For instance, another way to arrive at a similar scaling law would be to choose a different modeling for the failure distribution, based perhaps on program length, and introducing the notion of a distribution of program lengths corresponding to different samples, similar to Ringel and de Bem (2018). In the end, this type of construction will have a similar interpretation in terms of task complexity w.r.t the model.

We believe our toy model offers a simple yet effective phenomenological framework for understanding how inference quality improves with more opportunities to predict correctly. Future work could extend this framework to more complex models, including applying similar methodology as (Maloney et al., 2022) to generalized linear regression, kernel regression and neural networks, and investigate how it interacts with existing scaling laws based on model size and training data.

## Acknowledgements

We thank Yohai Bar-Sinai, Alon Beck, Itay Lavie, Nadav Outmezguine, Zohar Ringel and Antonio Sclocchi for fruitful discussions. The work of NL is supported by the EPFL AI4science program.

## Impact Statement

This paper presents work whose goal is to advance the field of Machine Learning. There are many potential societal consequences of our work, in particular due to the large model sizes considered in this work, but we do not feel there are specific aspects of this work with broader impacts beyond the considerations relevant to all large machine learning models.

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

## A. Experimental Setup

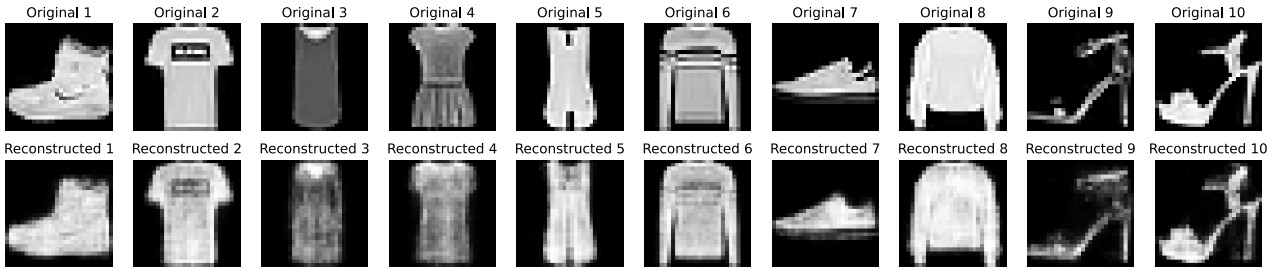

*Figure 8.* **Examples of original and reconstructed samples from the VAE reconstruction task.**

Here, we specify the experimental procedure used for the analysis in Section 5.

We utilize a Variational Autoencoder (VAE) with the following architectural details:

- **Input dimension:** $28 \times 28 = 784$, corresponding to the flattened pixel values of the Fashion MNIST dataset.

- **Hidden dimension:** 400.

- **Latent dimension:** 20, controlling the bottleneck for information in the latent space.

- **Decoder:** The decoder reconstructs the original input through two fully connected layers, outputting a 784-dimensional vector followed by a sigmoid activation to ensure pixel values remain between 0 and 1.

- **Temperature parameter:** A temperature parameter $T = 1.1$ is applied during the reparameterization step to control the variance of the latent variables, allowing us to model uncertainty in the latent space more effectively.

The VAE was trained on the first 400 samples from the Fashion MNIST dataset. The loss function combines binary cross-entropy for reconstruction and the Kullback-Leibler divergence to regularize the latent variables. We ran the training for 1000 epochs using the Adam optimizer with a learning rate of $1 \times 10^{-3}$.

To provide qualitative insights into the model's performance, we visualize several input samples from the Fashion-MNIST dataset along with their corresponding reconstructions in Figure 8. This allows us to inspect both successful and failed reconstructions, and examine the types of errors the model makes.

## B. Example of a Memorizing and Inferring Model

This appendix details the experimental setup used to simulate a neural network as a "memory" system with sample-specific retrieval probabilities, and its evaluation using a pass@k metric.

### B.1. Dataset and Preprocessing

We utilized a subset of the MNIST dataset (Deng, 2012).

- **Subset Size**: We selected the first $N_{samples} = 1000$ images from the MNIST training set.

- **Unique Class Assignment**: Each of these $N_{samples}$ images was treated as belonging to its own unique class. Thus, the classification task for the memory model involved $N_{samples}$ distinct classes. The label for the $i$-th sample in the subset was simply $i$.

- **Image Preprocessing**: MNIST images, originally $28 \times 28$ pixels, were flattened into vectors of size $D_{in} = 784$. Pixel values were normalized using the standard MNIST mean (0.1307) and standard deviation (0.3081).

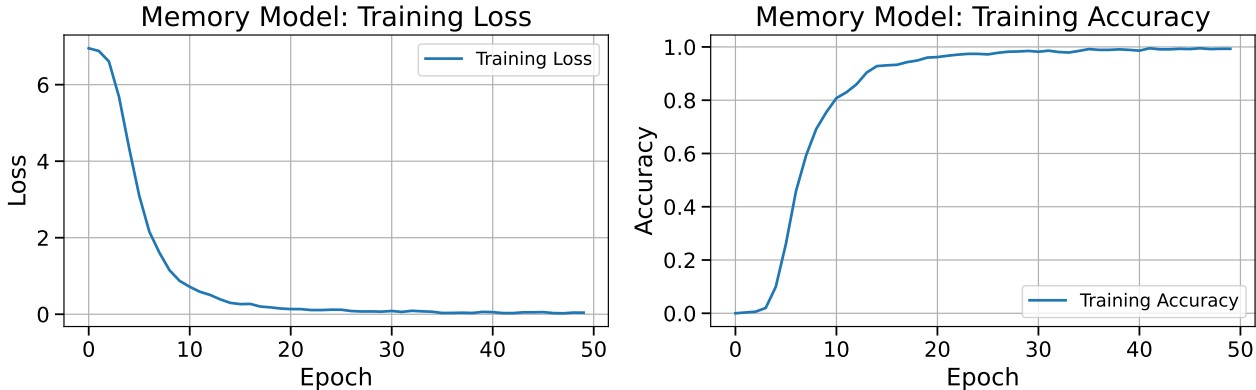

*Figure 9.* **Training curves for the "memory" model in Appendix B.**

## B.2. Memory Model

The "memory" component was implemented as a simple Multi-Layer Perceptron (MLP).

- **Architecture**: The MLP consisted of:
    1. An input layer accepting $D_{in} = 784$ features.
    2. A first hidden layer with 256 neurons, followed by a ReLU activation function.
    3. A second hidden layer with 128 neurons, followed by a ReLU activation function.
    4. An output layer with $N_{samples}$ neurons (one for each unique class), producing logits.

- **Training Objective**: The model was trained to perform classification, mapping each unique input sample to its assigned unique class index. The goal was for the model to effectively memorize its training data.

- **Training Parameters**:
    - **Loss Function**: Cross-Entropy Loss ($\mathcal{L}_{CE}$).
    - **Optimizer**: Adam optimizer (Kingma and Ba, 2017).
    - **Learning Rate**: $\eta = 0.001$.
    - **Epochs**: The model was trained for $E = 50$ epochs.
    - **Batch Size**: $B = 32$.

- **Training Monitoring**: Training loss and accuracy on the training set (which comprised all $N_{samples}$ unique items) were monitored per epoch. The expectation was for the model to achieve near-perfect accuracy, indicating successful memorization, as shown in Figure 9.

## B.3. Probabilistic Retrieval Process with Sample-Specific Difficulty

To simulate a retrieval process where samples have inherent difficulties, we assigned a success probability to each sample. This approach differs from directly adding noise to the model's output logits.

- **Sample Failure Probability Assignment**: For each of the $N_{samples}$ unique training samples $s$, an intrinsic probability of failing recall in a single trial, $p_s$, was drawn from a Beta distribution:

$$p_s \sim \text{Beta}(\alpha_S, \beta_S)$$

The parameters for this Beta distribution were set to $\alpha_S = 0.3$ and $\beta_S = 0.5$. This ensures that some samples are inherently "easier" (lower $p_s$) or "harder" (higher $p_s$) to retrieve. This $p_s$ is fixed for sample $s$ across all its retrieval trials, as shown in Figure 10.

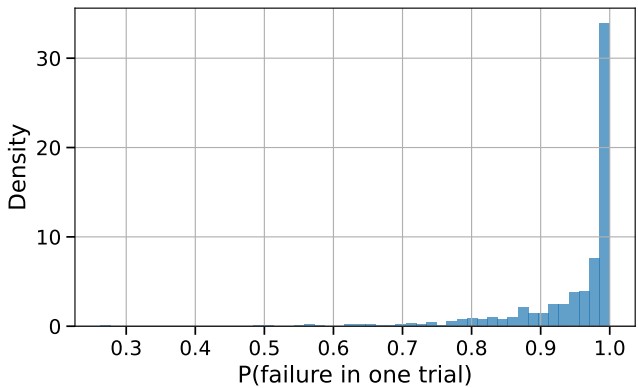

*Figure 10.* **Failure probability distribution for the inference model in Appendix B.**

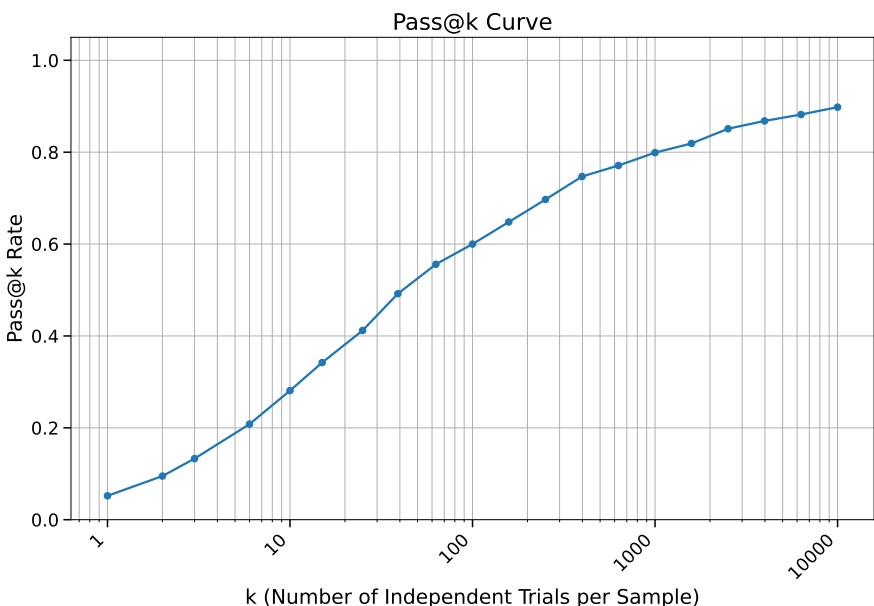

*Figure 11.* **Pass@k curve for the inference model in Appendix B.**

- **Simulating a Retrieval Trial**: For a given sample $s$ with its associated failure probability $p_s$, a single retrieval trial was simulated as follows:

  1. A random number $u$ was drawn from a uniform distribution, $u \sim \mathcal{U}(0, 1)$.
  2. If $u > p_s$, the trial was considered a "successful recall." In this event, it was assumed that the memory model would correctly identify the sample's true class as its top-1 prediction.
  3. If $u \leq p_s$, the trial was considered a "failed recall."

### B.4. Evaluation Metric: Pass@k

The performance of the probabilistic retrieval process was evaluated using the Pass@k metric.

- **Definition**: For a given sample $s$, Pass@k evaluates whether the true class of $s$ is successfully recalled (as defined above) in at least one of $k$ independent retrieval trials.

- **Calculation**:

1. For each sample $s$ in the dataset:
   – Up to $K_{max} = 10^4$ independent retrieval trials were simulated.
   – If a successful recall occurred at trial $t \leq k$, the sample was marked as passing for all $k' \geq t$ (up to $K_{max}$) for which pass@k' was being evaluated. The simulation for sample $s$ could then stop.
2. The overall Pass@k rate for a given $k$ is the fraction of the $N_{samples}$ samples that passed at $k$ trials.

The results are shown in Figure 11, with the precise phenomenology captured by the main text.

