# OpenReview forum: "A Simple Model of Inference Scaling Laws"
_ICML.cc/2025/Conference — ICML 2025 poster_

### Official Review · Reviewer_i2X4 · 2025-03-03

**Overall Recommendation:** 3

**Summary:**

This paper investigates how neural models' inference performance scales with multiple attempts, particularly in the context of LLMs. The study introduces a straightforward statistical framework based on memorization to explore the relationship between inference attempts and success rates, measured by the coverage or pass@k metric. This metric reflects the probability of obtaining the correct answer across repeated attempts. The authors derive an "inference loss" that shows a predictable decay in error with increasing trials, linking it to prompting costs. They validate their model through empirical experiments with LLMs on reasoning tasks and a generative VAE model, confirming that their theoretical predictions align with observed data. The framework isolates the effects of inference scaling and proposes it as a foundational element for optimizing the trade-off between training and inference costs to enhance overall model performance.

**Claims And Evidence:**

Some of the claims are not supported. Please refer to the questions for more details.

**Essential References Not Discussed:**

I think all important references are included.

**Experimental Designs Or Analyses:**

The existing experiments in this paper are sound, but some of the claims are not supported by the experiments. Please refer to the questions for more details.

**Methods And Evaluation Criteria:**

The proposed method and evaluation appear to make sense.

**Other Comments Or Suggestions:**

Some typos:

1. In Eq. (1), $E_i$ is not defined;

2. In line 305, "asmyptotes" should be "asymptotes".

**Other Strengths And Weaknesses:**

**Strengths**:

1. The topic is timely and aligns with the current research trend in reasoning models. Pass@k is an important metric in these models, particularly for MCTS or exploration in RL.

2. The proposed model is simple and elegant.

3. It is encouraging to see that the proposed model performs well in the experiments, as evidenced by Figures 1(a), 2, 6(a), and 7(b).

**Weaknesses**:

I have several concerns about the proposed model:

1. I have several questions regarding the practical aspects of data generation models. (Please refer to questions 1-4)

2. Some of the experiments do not align well with the scaling laws. (Please refer to questions 5 and 6)

**Questions For Authors:**

1. This paper focuses on the inference scaling laws of LLMs. However, the data generation model appears to be not relevant to LLMs. I suggest that the authors discuss why the practical problems in LLMs that require inference scaling laws should follow the Hutter model.

2. It would be interesting to investigate whether the proposed model is better than a simple power law scaling. An alternative model could be $pass@k = \mathcal{A} \times (\text{Sigmoid}(\alpha k) + \beta)$. Can the proposed model fit the curve better than this simpler model?

3. To fit Figure 1(a) with Equation (7) and Figure 2 with Equation (9), should we use the same set of parameters ($\mathcal{A}, \alpha, \beta$) or different sets? According to the model, the same set of these numbers should be used given a model.

4. In Figure 1(b), it is unclear how well the Beta distribution fits the practical data. The same issue arises for Figure 4(c).

5. In Figures 6(b) and 7(a), only a small proportion of the data fits the theoretical curve well. Why does this happen?

6. In Figure 6(a), the "emergent behavior" occurs when $\epsilon=0.1$, but not for other values. Why is this the case?

**Relation To Broader Scientific Literature:**

This problem is timely, as o1 and r1 are among the most popular research topics recently. The proposed model is simple and elegant. If the authors can address the concerns for this model, I believe it will inspire future reasoning models.

**Theoretical Claims:**

I ensure that the proof logic is correct.

---

> ### Author Rebuttal · Authors · 2025-03-30
>
> Dear Reviewer i2X4,
>
> Thank you for your careful and positive review of our paper, finding that our proposed model is simple and elegant. We hope that our responses will alleviate your concerns.
>
> Below, we address the issues you raised.
>
> **Comments and Suggestions**
>
> 1)	Thank you for pointing this out, we will complete L87: *For $n$ training samples, the expected single feature error $E_i$ is*.
> 2)	Thank you for finding this typo, it will be fixed in the revised version.
>
> **Weaknesses  and Questions**
>
> 1) *Data generation process* - While it is true that the data generation process of LLMs may seem very different from the simple memorizing model,  the claim of the paper is not that the simple model perfectly captures every detail of the LLM inference process, but rather the opposite – that these details are not crucial to address inference scaling, as it is often true with pre-training scaling laws [1,2].
> The only requirements from the LLM in order to fit our assumptions are that the model is sufficiently large, or has sufficient capacity to (approximately) store its training data, and that inference is done by an imperfect sampling process. The specific task we chose – which is recovering exactly the training data itself, is not what LLMs do, but it is a proxy for the task of “recovering the right answer from the training data”. Meaning that if the answer, or the path to the answer (a sequence of retrievals from its memory) existed in the training data, this model is able, with some error probability, to reach this answer. This approach explains why this simple model captures very different scenarios (LLM and VAE). In order to understand the precise connection between our model and LLM performance, it would be wise to study the internal representations of LLMs, and look for “memorizing modules”, possibly in specialized attention heads, as was suggested by reviewer kFgD, and will be part of the focus of our future works. We will better explain this point in the main text of the revised version, if the reviewer finds this answer acceptable and useful in understanding our goal.
>
> 2)	*Other functional fits to the data* - We agree with the reviewer that ablation with simple functions could be useful, we will include a short appendix considering several functions of the suggested type to show that they cannot be used to capture the full behavior of the model, typically sigmoid type functions are somewhat able to capture the small $k$ regime, but never both limits. We provide a preliminary  figure in https://imgur.com/kTsDM0R .
>
> 3)	*Similar parameters for different descriptions* - The parameters should not be the same, but there exists a simple mapping between them that can be derived from the large and small $k$ limits.
>
> 4)	*Beta fit to practical data* - Thank you for this comment. The two figures are conceptually different. Regarding Fig  4(c), we will include the empirical curves, which match very well the asymptotic behavior predicted by our model (https://imgur.com/a/8Ftb76R). Regarding Fig 1(b), this is a prediction made by our model that requires further study to properly interpret. In essence, this figure shows the "difficulty" distribution for the different models, on the specific set of maths questions. It does not fit any empirical data, but should be taken as the interpretation of the $\alpha,\beta$ parameters. The goal of the figure is to show that different models can perceive the same data at different difficulty levels, and a detailed study of this point could lead to new inference sampling techniques, based on the ratios of easy and difficult questions. We hope this explanation clarifies this point and would be happy to discuss further.
>
> 5)	*Figures 6(b) and 7(a) fitting the theoretical curve* - Figs 6(b) and 7(a) are meant to serve as qualitative evidence for the effective correlation between trials, rather than an exact characterization. A priori, the inference attempts themselves need not be correlated, but we see that there are effective long range correlations between different trials from both the reconstruction error 6(b) and its eigenvalues 7(a). We see that it is sufficient that only a bulk of the eigenvalues conform to a power-law decay to have good predictions for the pass@k metric. We will explain this point more clearly in the revised text.
>
> 6)	*emergent behavior at specific threshold value* -  The logic is that for larger threshold values the model must have smaller reconstruction error, and so require fewer inference attempts to succeed. The "emergence" is just a feature of the functional form of Eq. 7. It would be interesting to analyze this model in more detail and consider whether one can define "emergence" as the point where the function changes from concave to convex perhaps, and study the internal structure of the model at these points.
>
> **References:**
>
> [1]  Maloney et al. https://arxiv.org/abs/2210.16859
>
> [2] Bahri et al. https://www.pnas.org/doi/abs/10.1073/pnas.2311878121

---

> > ### Comment · Reviewer_i2X4 · 2025-04-02
> >
> > My concerns have been addressed and thus I keep the positive score. For rebuttal 2, I think there are many other simple models in statistical mechanics and I recommend the authors to try them.

---

### Official Review · Reviewer_FFcT · 2025-03-08

**Overall Recommendation:** 3

**Summary:**

The paper introduces a statistical framework to analyze the scaling laws for LLMs inference, particularly addressing how model performance (pass@k) improves with repeated inference attempts. The authors present 2 models, one assumes samples differ in difficulty, modeled via a Beta distribution; the other considers correlated inference attempts through a power-law correlation struction. Empirical validation is conducted on large language models (LLMs) and a Variational Autoencoder (VAE) trained on image reconstruction tasks, showing strong agreement between theoretical predictions and empirical data.

**Claims And Evidence:**

The claims are supported by empirical experiments. The authors demonstrate that their proposed analytical framework for inference scaling match the empirical "pass@k" curves observed in multiple LLM experiments and VAE reconstructions. The findings indicate that inference performance improves predictably with repeated attempts.

**Essential References Not Discussed:**

I am not sure about the essential references not discussed.

**Experimental Designs Or Analyses:**

Experimental designs for both the LLM-based mathematical tasks and the VAE reconstruction tasks are sound and appropriate.

**Methods And Evaluation Criteria:**

The proposed methods are reasonable and suitable for capturing the inference scaling phenomena in LLMs and generative models. pass@k is a reasonable evaluation criteria for measuring model's performance in math reasoning and coding problems.

**Other Comments Or Suggestions:**

N/A

**Other Strengths And Weaknesses:**

Strengths: The problem this work is trying to address is of practical significance. The findings offer valuable insights into practical strategies for balancing computational costs and model performance by adjusting inference attempts.
Weakness:
- The strong assumptions regarding independence and perfect verification limit the model's direct applicability in less controlled scenarios.
- Using more diverse models (other than small VAE) could make the findings more persuasive.

**Questions For Authors:**

Could you clarify if your scaling law predictions hold when applied to tasks beyond mathematical reasoning or VAE reconstruction, especially for the tasks that cannot be verified easily.
How sensitive are your conclusions to the choice of parameters α and β in the Beta distribution? How will different choices of these parameters affect the estimation?

**Relation To Broader Scientific Literature:**

I am not sure about its relation to broader scientific literature.

**Theoretical Claims:**

The paper provides analytical derivations for the inference scaling behavior under specific assumptions (e.g., independence and correlation of trials). The correctness of these derivations looks mathematically sound.

---

> ### Author Rebuttal · Authors · 2025-03-30
>
> Dear Reviewer FFcT,
>
> Thank you for your positive appraisal of our submission. We’re glad that you found valuable insights and potential practical applications, which were our goals.
>
> We address the weaknesses raised, as well as questions below.
>
> **Weaknesses**
>
> 1)	*strong assumptions* - We completely agree that both the independence assumption and the perfect verification limit are strong assumptions. However, given that our paper is the **first** to propose a model for inference scaling (as far as we know), it is natural to begin with a limiting, ideal setting and extend it in future works. It is rather surprising that even under these strong assumptions, the simple memorizing model captures the inference scaling behavior of real models as well as it does.
>
>    We would like to kindly bring to the reviewer’s attention the fact that the independence assumption itself is also discussed in the main text, and the “effectively correlated trials” section (Sec. 4.2) is meant to describe this effect, if we understand the reviewer correctly. We apologize if this point was not sufficiently clear, and will try to highlight it further in the revised version.
>    Furthermore, we are sure that relaxing the perfect verification assumption will be a very interesting avenue for future works, since verification methods is a very broad topic all in itself.
>
> 2)	*More diverse models* - We agree in principle that adding more diverse controlled experiments could extend the scope of the paper, but we believe that the current evidence on LLMs and VAE reconstruction supports our predictions. The VAE experiments are meant as intermediate step, that shares some of the complexity of LLMs without the full pipeline. If the reviewer has a particular experiment in mind that would make sense within the “memorization-inference” setup, which is neither VAE nor the LLMs, we would be happy to test our predictions on it.
>
> **Questions**
>
> 1)	*Scaling beyond tasks in the main text* - The experiments given in the main text which consider LLMs on reasoning tasks and the VAE reconstruction are different, but share some common features. In particular, in both tasks, the model is asked to “learn” the training data distribution and perform sampling (as opposed to regression for instance). In that sense, the exact details of the architecture, model and task are not particularly important, as long as the model can be thought of as a two component “memorization” module and  “inference” module. Therefore, our predictions are quite universal, as long as the data is not uniformly “easy”, and so we believe our results should extend to other generative settings, for instance to diffusion models.
>
> 2)	*Sensitivity to $\alpha,\beta$* - The question of sensitivity to the parameter choices is a bit unclear to us, since the conclusions (namely the inference scaling laws) are analytically given in terms of $\alpha,\beta$, so the exact dependence on the parameters can be characterized. Could the reviewer please clarify their meaning? If the question is of interpretation, then different choices of $\alpha$ correspond to a different small $k$ behavior, since as we explain in the main text, the average “difficulty” of the samples is $\frac{\alpha}{\alpha+\beta}$, and so larger $\alpha$ values would imply a slower improvement with $k$ for a smaller number  of trials, while $\beta$ dictates the large $k$ behavior of the inference loss/pass@k, meaning how difficult it is to improve performance by increasing $k$ at a large number of inference attempts, where larger $\beta$ means greater scaling improvement.
>
> We hope that our replies are sufficient to raise the reviewer’s confidence in our work, and potentially accept the paper. We welcome any further questions and comments.

---

### Official Review · Reviewer_kFgD · 2025-03-11

**Overall Recommendation:** 3

**Summary:**

This paper studies the paradigm of inference scaling and tries to identify the functional form that can help explain predict performance therein, i.e., building a scaling law with respect to inference budget (e.g., k in pass@k metric). The authors consider one of the simplest models for pretraining scaling laws from literature, proposed by Hutter (2021) (and originally dating back to several people in 1990s, including Amari), that considers a hypothesis class wherein the data has been perfectly memorized by the network. Assuming this perfect memorization, the rate of loss reduction can be shown to yield a power law under certain assumptions on the data distribution. The authors essentially extend this model to inference scaling, finding really good fits to the empirical results.

**Claims And Evidence:**

The claims are well-backed with evidence: the empirics are thorough and consistency with theory is really nice. I really liked the VAE results---they were honestly the most exciting confirmation of the model and I'd have loved to see them emphasized more in the writeup.

**Essential References Not Discussed:**

Related work is fairly described in my opinion.

**Experimental Designs Or Analyses:**

Yes, see above.

**Methods And Evaluation Criteria:**

Both approaches and results make sense, and I do not have any complaints.

**Other Comments Or Suggestions:**

Implementation: While the authors motivate their model of scaling with the memorization ansatz that derives from Hutter's work, it would have been better to see a more mechanistic argument for why we should expect such a model to transpire for a pretrained Transformer model. Is it possible the results are a consequence of specialized attention heads basically operating like perfect memorization modules? If not, then would we expect the scaling to be very different for a different architecture such that the proposed memorization model breaks down there? For ex., would inference scaling with an RNN not work. (I realize inference scaling with RNNs has not been shown, but I'm mostly asking for curiosity's sake here.)

**Other Strengths And Weaknesses:**

See above.

**Questions For Authors:**

See above.

**Relation To Broader Scientific Literature:**

To my knowledge, this paper offered the first theoretical model for inference scaling. A very loosely related paper that comes to mind is by Park et al. (https://arxiv.org/abs/2501.00070), which in fact shows sample complexity of inference scaling for a belief update task (unlike the ones considered in this paper), is worse than a memorization based model would suggest, but nevertheless highly predictable. I'd actually be curious to hear authors' thoughts on that paper.

**Theoretical Claims:**

Not quite applicable, since the model is an extension with fairly reasonable and well-motivated assumptions on the fitting parameters.

---

> ### Author Rebuttal · Authors · 2025-03-29
>
> Dear Reviewer kFgD,
> Thank you for carefully reading our manuscript, we are glad that you found our theoretical analysis sound and our empirical results well backed.
>
> We would like to address the various points you raised below.
>
> **Regarding Evidence**
>
> We agree that the VAE results are interesting, and are happy to see that they were appreciated. To be honest, having discussed with other researchers and due to the broader interests of the community, we preferred to highlight the functional matching to the LLM performance rather than the more controlled VAE setting. If you believe this is sufficiently interesting to the community, we will try to highlight these results a bit more in the revised manuscript.
>
>  **Regarding Broader Scientific Literature**
>
> As far as we know, this is indeed the first theoretical model for inference scaling. From a brief reading of the Park et al. paper, we agree that there are interesting connections there, though their work is for in-context learning and ours was tested on inference without further context. It is not unreasonable to think that one way to interpret the in-context update to the representation as somewhat equivalent to resampling from the memory of the pre-trained model during inference. Perhaps in the ICLR setting our "error probabilities" are not fixed, but updated according to the context provided. We thank the reviewer for bringing this to our attention, and will add it to the related works. If the reviewer has more insights to share along these lines during the discussion period, possibly regarding something we've missed, we'd be happy to discuss it further.
>
> **Comments/Suggestions**
>
> This is an extremely interesting question, and we agree that developing a mechanistic understanding of the memorization module ansatz would be a valuable complement to our simple model. Our goal in this work was to introduce a first model that provides a clear explanation of the key inference observation. Understanding how this final result emerges from training would be a natural extension.
>
> As a next step—currently a work in progress—we aim to explore the connection between pre-training scaling performance and inference scaling in solvable models, which we believe exist. This direction may align with the reviewer's intuition, as memorization must occur during training, and we may observe different inference scaling behaviors depending on whether the "memorization assumption" holds or breaks down. Moreover, the simple model presented here seems to give quite universal results, but it still might be that there are several "universality classes", where different models might fall into.
>
> Finally, while we have not yet considered RNNs, their study should, in principle, be feasible, and would very likely lead to different scaling behaviors, at least in some cases (if not considering SSMs perhaps).
>
> We appreciate this valuable suggestion and will certainly investigate it in future work.
>
> We hope that our replies have affirmed your confidence in our work, and potentially lead to accepting the paper. We welcome any further questions and comments.

---

### Official Review · Reviewer_4E5M · 2025-03-13

**Overall Recommendation:** 3

**Summary:**

The paper proposes to study scaling laws for inference in a restricted setup where the model can potentially memorize the training dataset. The paper also shows that the theoretical predictions match empirical results on mathematical reasoning tasks for LLMs.

**Claims And Evidence:**

The paper makes several assumptions about the setup (e.g. the model can memorize all the samples upto the model capacity (line 101 column2) or that the model makes an incorrect prediction with some probability or that Beta distribution can be used to model the distribution of difficulty of data points. Using these assumptions, the paper derives scaling laws and the theoretical results are shown to match the empirical results.

**Essential References Not Discussed:**

I am not much familiar with this literature but the paper seems to cite relevant related work.

**Experimental Designs Or Analyses:**

Given the focus on theoretical results, the experiments, while not exhaustive, seem sufficient to back up the main claims.

**Methods And Evaluation Criteria:**

The paper largely focuses on theoretical results and while they could have used more complex dataset for their empirical results, it should be fine given the focus on theory.

**Other Comments Or Suggestions:**

1. The authors should include some motivational real-world examples where the memorization assumption holds (in approximation)

2. In line 425 (second column), "Similar to ( Ringer..)" should be "Similar to Ringer.. "

**Other Strengths And Weaknesses:**

I found the paper (and some captions) to be a bit dense to understand. e.g. The caption for Figure 1 was quite dense. I found the experiments with Llama-3-8B to be well motivated and easy to follow.The paper makes both theoretical and empirical contributions and should be useful for the community

**Questions For Authors:**

1. In section 4, why is beta distribution a good choice ?

**Relation To Broader Scientific Literature:**

I am not much familiar with this literature but the paper seems to build on previous works, in terms of scaling law analysis and the choice of the memorization setup. It also improves on the existing work by focusing more on the inference scaling laws and incorporating the number of inference steps in their analysis.

**Theoretical Claims:**

I do have some questions about assumptions that the authors make (more of this in "Questions For Authors"). I could broadly follow the theoretical claims but I did not attempt to re-derive final expressions in different equations (e.g. equation 7, equation 9, equation 10). I should also say that I will rely quite heavily on the other reviewers to fully appreciate the theoretical contributions (as this is not my regular area of work) so the authors should focus on addressing their questions and concerns first.

---

> ### Author Rebuttal · Authors · 2025-03-29
>
> Dear Reviewer 4E5M,
> Thank you for your positive reading of our manuscript.
>
> Below, we address your comments/questions:
>
> **Weakness:**
>
>   *The caption for Figure 1 was quite dense* – In the revised version, we will shorten the caption, and move some of its content to the main text, namely L131 – 133 could be moved. Do you believe this will make the caption easier to understand?
>
> **Comments/Questions:**
>
> 1)	*The authors should include some motivational real-world examples where the memorization assumption holds (in approximation)* - As per your suggestion, we will include an appendix which contains a very simple neural network classifier which can memorize (approximately) its training data, combined with an inference model which is allowed to make mistakes based on the Beta distribution. This way we separate the memorizing assumption from the inference assumption.
> 2)	Thank you for the correction, we will fix the mistake.
>
> **Questions:**
>
> 1)	Regarding the Beta distribution - The Beta distribution is a particular choice made in this paper, but it is certainly not unique, as we point out in the main text: *One way to model…*. The reason for this choice is the fact that the inference behavior differs for small $k$ and large $k$, and so at least a two-parameter distribution is required. For instance, using the classical Zipf law type distribution with a single decay parameter $\alpha$ would only capture one of these limits. Other two parameter distributions would also be acceptable, but the interpretation will remain the same: the model perceives some inference tasks as “difficult” and others as “easier” depending on the two parameters of the distribution.
>
> We hope that our replies are sufficient to raise the reviewer’s confidence in our work, and potentially accept the paper. We welcome any further questions and comments.

---

### Decision · Program_Chairs · 2025-05-01

**Decision:**

Accept (poster)

**Comment:**

This paper proposes a simple theoretical model of inference-time scaling laws, with accompanying empirical validation. Overall, I agree with the reviewers' assessment: neither the experiments nor the theory are particularly groundbreaking on their own, but it is refreshing to see a simple, understandable theoretical model whose predictions are actually reflective of experimental practice. I recommend accepting this paper.